# Tire Model with Temperature Effects for Formula SAE Vehicle

**Diwakar Harsh** [1] **and Barys Shyrokau** [2],* 

[1] Rimac Automobili d.o.o., 10431 Sveta Nedelja, Croatia; diwakar.harsh@rimac-automobili.com
[2] Department of Cognitive Robotics, Delft University of Technology, 2628CD Delft, The Netherlands
* Correspondence: b.shyrokau@tudelft.nl

**Abstract:** Formula Society of Automotive Engineers (SAE) (FSAE) is a student design competition organized by SAE International (previously known as the Society of Automotive Engineers, SAE). Commonly, the student team performs a lap simulation as a point mass, bicycle or planar model of vehicle dynamics allow for the design of a top-level concept of the FSAE vehicle. However, to design different FSAE components, a full vehicle simulation is required including a comprehensive tire model. In the proposed study, the different tires of a FSAE vehicle were tested at a track to parametrize the tire based on the empirical approach commonly known as the magic formula. A thermal tire model was proposed to describe the tread, carcass, and inflation gas temperatures. The magic formula was modified to incorporate the temperature effect on the force capability of a FSAE tire to achieve higher accuracy in the simulation environment. Considering the model validation, the several maneuvers, typical for FSAE competitions, were performed. A skidpad and full lap maneuvers were chosen to simulate steady-state and transient behavior of the FSAE vehicle. The full vehicle simulation results demonstrated a high correlation to the measurement data for steady-state maneuvers and limited accuracy in highly dynamic driving. In addition, the results show that neglecting temperature in the tire model results in higher root mean square error (RMSE) of lateral acceleration and yaw rate.

**Keywords:** tire model; tire temperature; FSAE vehicle

## 1. Introduction

Formula SAE (FSAE) also known as formula student (FS) is a competition in which students design a single seat formula race car to compete against other FS teams from all over the world [1]. The competition motivates the students for innovative solutions to demonstrate their engineering talent and obtain new skills. It also allows the students to apply theoretical knowledge into practice in a dynamic and competitive environment.

Since 2001 students have joined Formula Student Team Delft [2] on an annual basis to participate in the FSAE competition, earlier in the combustion class and electric class from 2011. Aiming to improve the FSAE vehicle performance, the team collaborated with Apollo Vredestein B.V. (Enschede, The Netherlands) to develop original tires. The FSAE vehicle (2017 version) from FS Team Delft equipped with such tires is shown in Figure 1.

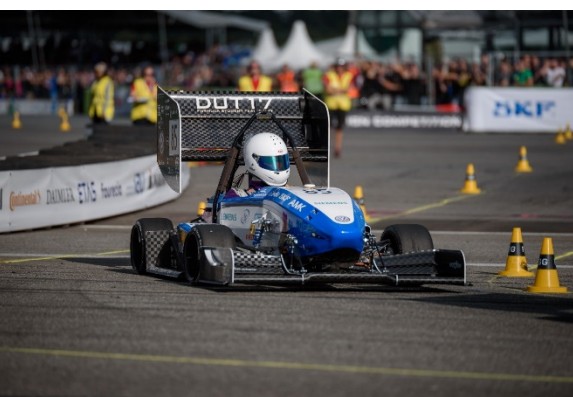

**Figure 1.** DUT17 Formula Society of Automotive Engineers (FSAE) vehicle at the Formula Student Germany competition [2].

To achieve the best FSAE vehicle performance, the team developed various tools to evaluate vehicle dynamics and to predict its behavior. However, predictions are based on estimated friction coefficient, neglecting the thermal effect on the tire performance. Frequently, lap simulation is based on a point mass or simplified vehicle models allowing for the development of the top-level concept of the new vehicle. It can quantify vehicle parameters such as the height of the center of gravity and lift coefficient. However, to design various vehicle components and, specifically, control algorithms, a full vehicle simulation including a comprehensive tire model is required.

For an accurate modeling of tire forces and moments, a physical, semi- or empirical tire model can be used [3]. The physical approach provides more insights regarding tire behavior and better represents the whole operation range; however, it lacks accuracy, specifically considering camber influence, conicity, plysteer, and other phenomena [4,5]. The empirical approach provides a higher modeling accuracy in the predefined tested range but requires extensive special tests. According to [6,7] the tire temperature, especially for racing application [8], has a significant effect on the force capability of the tire. Several advanced tire models using the brush element approach incorporated with thermal effects have been proposed [9–11]; however, their parametrization requires intensive test sessions, commonly unfeasible for FS teams. For a FSAE vehicle, less complex models incorporating temperature effect are proposed using the physical approach based on the brush model [12,13]. They demonstrate a good correlation with the experimental tire data collected using the indoor flat track tire test machine [14]. Also, the proposed physical-based models were compared to the original magic formula demonstrating close accuracy; however, original magic formula does not include the effect of temperature on the force capability of the tire.

Thus, the goal of the proposed study was to develop an empirical tire model incorporating thermal effects and to evaluate its performance compared to the experimental measurements. The main contribution of the study is the improvement of the accuracy of the empirical tire model using the proposed thermal model and modification of the magic formula to incorporate temperature effects.

The paper is structured as follows. Section 2 describes the experimental setup, test program, and experimental results. A thermal tire model is proposed in Section 3 discussing the basic thermal equations and comparing them to well-established thermal models. Section 4 presents the widely used magic formula with the extension related to the temperature effect. The complete vehicle model including thermal and modified magic formula is presented in Section 5; it includes the whole model simulation for steady-state and transient maneuvers compared to real tests and simulation without the temperature effect. The paper concludes with a discussion of the work and the recommendations for future research presented in Section 6.

## 2. Experimental Setup and Results

### 2.1. Setup and Test Program

The experimental tests were performed at Dynamic Test Center AG [15] located in Vaufellin, Switzerland. The test program is shown in Table 1 covering uncombined slip testing using sweep tests of slip angle $\alpha$ and wheel slip $\kappa$ under various normal loads $F_z$ for three tires: Apollo Single Compound, Apollo Double Compound, and Hoosier.

**Table 1.** Tire test program.

| Measurement Type | $\alpha$ (rad) | $\kappa$ (–) | $F_z$ (N) |
|---|---|---|---|
| Pure cornering | sweep | 0 | 600/1000 |
| Pure acceleration/braking | 0 | sweep | |

The measurements were performed outdoor using a test truck (Figure 2). To measure tire forces and moments, a Kistler wheel force transducer [16] was used. The measurement accuracy of the wheel force transducer is 1% of the full scale. For the investigated load range, it resulted in approx. 5% absolute measurement error including crosstalk. The normal load during the test was controlled using hydraulic suspension. The target normal load was obtained by the adjustment of hydraulic pressure. The wheel was driven through the axle unit from the motor for acceleration tests and using a reverse gear for braking. Another actuator was mounted on the tie-rod in order to generate the tire slip angle.

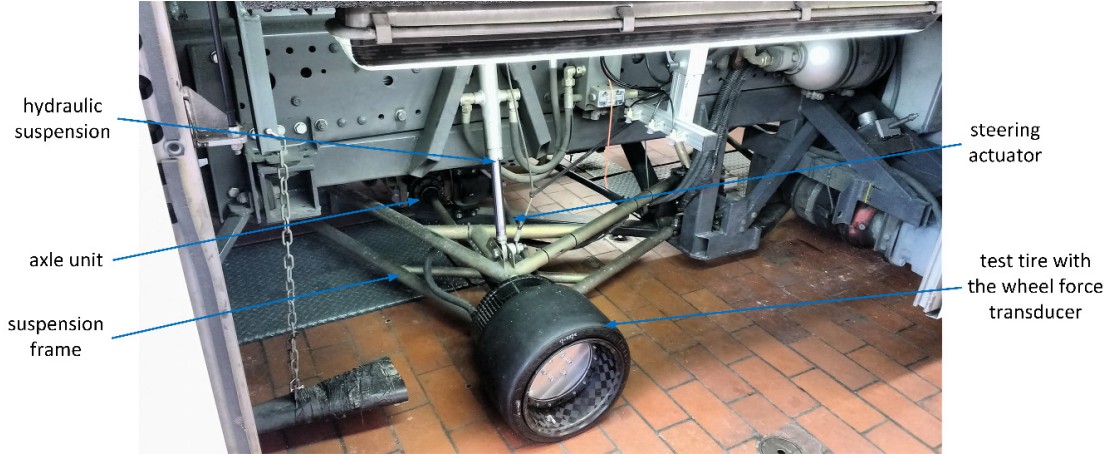

**Figure 2.** Test setup.

### 2.2. Pure Cornering Results

Tire lateral performance was assessed by performing sweep tests at a constant road wheel steering rate of 15°/s. The tests were conducted under various normal loads $F_z$ of 600 and 1000 N correspondingly. During the tests, the vertical force was fluctuated due to road modulation and other irregularities, which as a result affected the lateral force $F_y$ measurement. To obtain the lateral force, this influence was compensated using the normalized ratio between lateral and normal forces and then multiplying by mean vertical load. Figure 3a shows the effect of vertical load on the utilized friction coefficient $\mu$. The results are aligned to the load sensitivity phenomena that utilized friction coefficient reduces as normal load increases. The test results of Hoosier and Apollo tires (double compound) are compared in Figure 3b. The peak utilized friction for the Hoosier tire was around 5% higher than the Apollo one; however, the average utilized friction was in the same range.

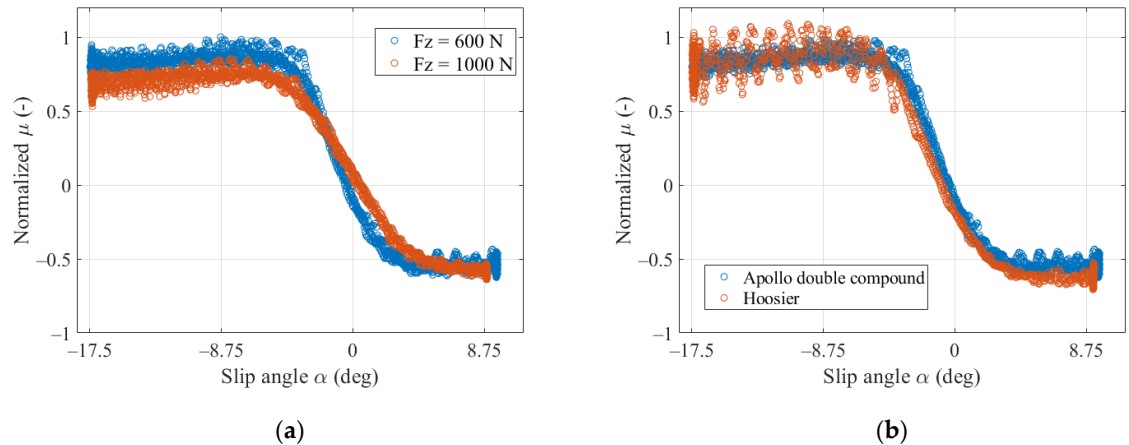

**Figure 3.** Normalized lateral friction $\mu$ vs. slip angle $\alpha$: (**a**) different normal loads $F_z$ (Apollo); (**b**) different tire types ($F_z = 600$ N).

### 2.3. Pure Acceleration and Braking Results

For the longitudinal force measurement, the wheel was lifted, accelerated, and then pressed onto the ground. The brake torque was applied to realize sweep tests of wheel slip while the forward velocity of the truck was kept constant. The effect of normal load on utilized friction is much less compared to lateral friction (Figure 4a). The performance of the Apollo tire was similar compared to the Hoosier one (Figure 4b). The difference was found to be 3% which can be related to the ambient and track conditions.

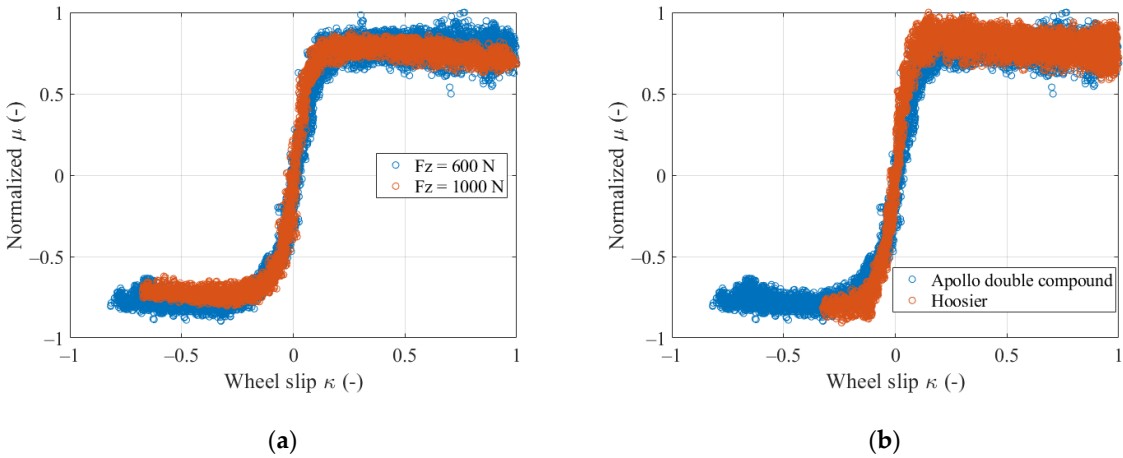

**Figure 4.** Normalized longitudinal friction $\mu$ vs. wheel slip $\kappa$: (**a**) different normal loads $F_z$ (Apollo); (**b**) different tire types ($F_z = 600$ N).

### 2.4. Results Related to Temperature Effect

Both the force capability and the tire lifetime depend on tire temperature. Since the tire carcass is an elastic element, the temperature change will result in the modulus of elasticity of the rubber and therefore influence the cornering stiffness [17]. The effect of temperature on the normalized longitudinal and lateral friction is shown in Figure 5. It can be observed that the temperature affects the stiffness (curve slope) and the peak friction.

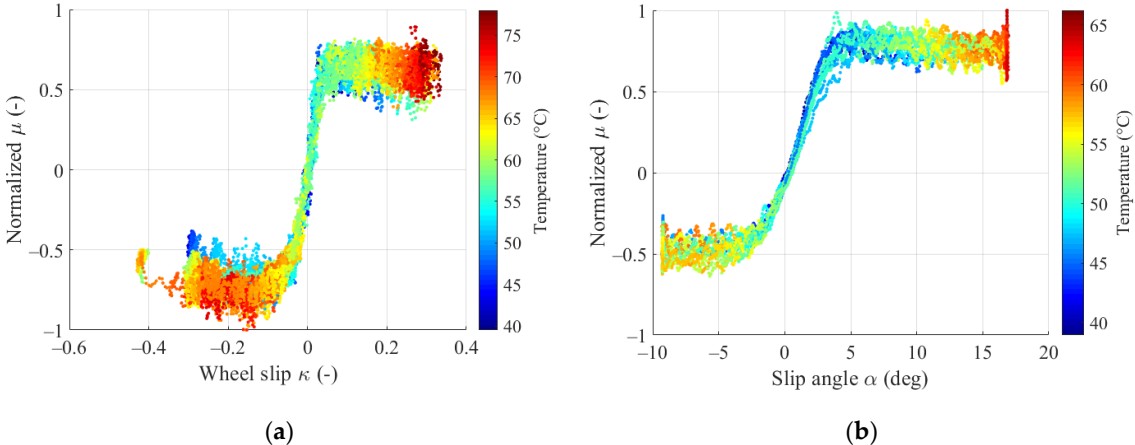

**Figure 5.** Temperature effects on tire forces for Apollo tire ($F_z$ = 1000 N): (**a**) normalized longitudinal friction $\mu$ vs. wheel slip $\kappa$; (**b**) normalized lateral friction $\mu$ vs. slip angle $\alpha$.

## 3. Thermal Model

### 3.1. Proposed Thermal Model

A lumped parameter approach is used according to [18] and the thermal model consists of three bodies such as the tread with temperature $T_{tread}$, the carcass with temperature $T_{carcass}$, and the inflation gas with temperature $T_{gas}$. The road surface with ambient air provides the boundary conditions with fixed temperatures $T_{road}$ and $T_{amb}$. In contrast to the study [19], a single value of tread temperature was calculated without the need to define a complex temperature distribution across the contact patch area. The temperature change of tread, carcass, and gas is described as:

$$
\begin{aligned}
\dot{T}_{tread} &= \frac{Q_{sliding} - Q_{tread \to road} + Q_{carcass \to tread} - Q_{tread \to amb}}{S_{tread} M_{tread}} \\
\dot{T}_{carcass} &= \frac{Q_{damp} - Q_{carcass \to tread} - Q_{carcass \to amb} - Q_{carcass \to gas}}{S_{carcass} M_{carcass}} \\
\dot{T}_{gas} &= \frac{Q_{carcass \to gas}}{S_{gas} M_{gas}}
\end{aligned}
\tag{1}
$$

where $Q_{sliding}$ is the heat flow due to sliding; $Q_{tread \to road}$ is the heat flow between tread and road; $Q_{carcass \to tread}$ is the heat flow between carcass and tread; $Q_{tread \to amb}$ is the heat flow between tread and ambient air; $Q_{damp}$ is the heat flow due to carcass deflection; $Q_{carcass \to amb}$ is the heat flow between carcass and ambient air; $Q_{carcass \to gas}$ is the heat flow between carcass and inflation gas; $S_{tread}$ is the specific heat capacity of carcass; $S_{carcass}$ is the specific heat capacity of carcass; $S_{gas}$ is the specific heat capacity of inflation gas; $M_{tread}$ is the tread mass; $M_{carcass}$ is the carcass mass; $M_{gas}$ is the inflation gas mass.

According to [18], two heat generation processes should be considered:

- Due to carcass deflection

$$
Q_{damp} = \left( E_x |F_x| + E_y |F_y| + E_z |F_z| \right) V_x
\tag{2}
$$

where $E_x$, $E_y$, and $E_z$ are the carcass longitudinal, lateral, and vertical force efficiency factors; $F_x$, $F_y$, and $F_z$ are the longitudinal, lateral, and normal forces; $V_x$ is the longitudinal velocity.

- Due to sliding friction in the contact patch

$$
Q_{sliding} = \mu_d F_z v_s
\tag{3}
$$

where $\mu_d$ is the dynamic friction coefficient; $v_s$ is the sliding velocity.

To define the shift along the $\mu$ axis with compound temperature, the dynamic friction model [18] was modified. Based on the friction model [20], Equation (4) is proposed incorporating the shift along

the $\mu$ axis due to temperature using the parameters $\mu$ and $h$ as temperature dependent. The idea was taken from research [21] where the parameters of the friction models from [20] and [22] were made temperature-dependent. The final equation for the dynamic friction model is described:

$$\mu_d(T) = \mu_{base} + \left[\mu_{peak}(T) - \mu_{base}\right]e^{-\left(h(T)\left(\log_{10}\left(\frac{v_s}{V_{max}}\right) - K_{shift}(T_{tread} - T_{ref})\right)\right)^2} \tag{4}$$

The temperature-dependent peak friction coefficient $\mu_{peak}$ is defined:

$$\mu_{peak}(T) = a_1 T^2 + a_2 T + a_3 \tag{5}$$

where $a_1$, $a_2$, and $a_3$ are the tuning parameters for a second order polynomial function.

Dimensionless parameter $h$ is related to the width of the speed range in which the friction coefficient varies significantly [23]. The following temperature-dependent adjustment is made similar to the adaptation of the compound shear modulus in [18]:

$$h(T) = b_1 \frac{e^{b_2 T_{tread}}}{e^{b_2 T_{ref}}} \tag{6}$$

where $b_1$ and $b_2$ are the tuning parameters.

The proposed modifications cover both sliding and non-sliding friction components. Temperature-dependent peak friction coefficient $\mu_{peak}$ corresponds to the peak friction properties of the tire compound. The parameter $h$ affects the curve slope and results in the change of the shear modulus of the tire compound. The peak friction coefficient $\mu_{peak}$, the lower limit of friction coefficient $\mu_{base}$, and the parameter $h$ are identified from measurement data. The simulation and experimental results for the obtained normalized lateral friction coefficient are shown in Figure 6. The model shows similar qualitative behavior observed in the experimental data at the slip angle above $4°$.

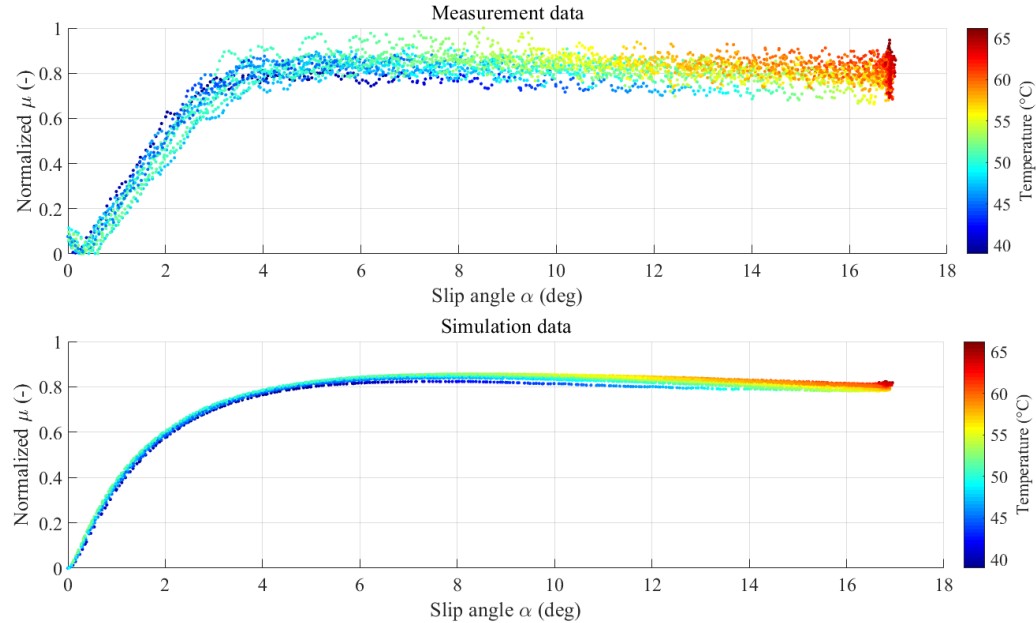

**Figure 6.** Comparison between measurement data and modified friction model.

Four types of heat flows are considered: (i) heat flow between tread and ambient air; (ii) heat flow between carcass and ambient air; (iii) heat flow between tread and carcass; (iv) heat flow between carcass and inflation gas.

Heat flow with the ambient air $Q_{tread\rightarrow amb}$ and $Q_{carcass\rightarrow amb}$ is defined as:

$$Q_{tread\rightarrow amb} = H_{tread-amb}(T_{tread} - T_{amb})$$
$$Q_{carcass\rightarrow amb} = H_{carcass-amb}\left(T_{carcass} - T_{gas}\right) \tag{7}$$

The heat flow coefficient $H_{carcass-amb}$ is constant and the heat flow coefficient $H_{tread-amb}$ is assumed to be a function of the longitudinal vehicle speed $V_x$ and taken as a linear function [18]:

$$H_{tread-amb} = 2V_x + 10 \tag{8}$$

Heat flow between the tread and carcass $Q_{carcass\rightarrow tread}$ is calculated as:

$$Q_{carcass\rightarrow tread} = H_{carcass-tread}(T_{carcass} - T_{tread}) \tag{9}$$

Heat flow between the carcass and inflation gas $Q_{carcass\rightarrow gas}$ is defined as:

$$Q_{carcass\rightarrow gas} = H_{carcass-gas}\left(T_{carcass} - T_{gas}\right) \tag{10}$$

Using heat flow coefficient $H_{tread-road}$, heat flow between tread and road $Q_{tread\rightarrow road}$ can be found as:

$$Q_{tread\rightarrow road} = H_{tread-road}A_{cp}(T_{tread} - T_{road}) \tag{11}$$

The contact patch area, $A_{cp}$, is calculated assuming a constant width of the contact patch $b$ and the variable contact patch length $a$ which was adjusted compared to [18] to better match the data of the FSAE tires:

$$A_{cp} = 2ab = 0.12p_{bar}^{-0.7}\left(\frac{F_z}{3000}\right)^{0.7}b \tag{12}$$

Since the volume of the gas is constant, the pressure is directly proportional to the Kelvin temperature and thus the inflation pressure $p_{bar}$ can be calculated as:

$$p_{bar} = p_{cold}\frac{T_{gas} + 273}{T_{amb} + 273} \tag{13}$$

where $p_{cold}$ is the pressure of the tire at ambient temperature.

*3.2. Comparison with the Established Models*

The Sorniotti model [12] describes an empirical model to estimate tire temperature as the function of the actual working conditions of the component. To evaluate the temperature effect on tire forces, a combination of the estimated temperature with a tire brush model [24] was used. The model was empirically tuned using experimental data to demonstrate the variation of tire performance as temperature function. The thermal model considers distinct thermal capacities related to the tread and carcass. The tread thermal capacity is related to the heat flux caused by the tire–road forces and carcass thermal capacity is affected by rolling resistance and exchanges heat with the external ambient. Other heat fluxes corresponded to the ambient and exchange between the two capacities. The original model [12] did not consider the heat flow between the road and tread. Therefore, the heat flow term $P_{tread,road}$ was introduced to the original model to improve the accuracy according to [25].

The model is described as:

$$C_{eq\_carcass}\frac{dT_{carcass}}{dt} = P_{rolling\_resis tan ce} + P_{conduction} + P_{ambient,carcass}$$
$$C_{eq\_tread}\frac{dT_{tread}}{dt} = P_{Fx,tire} + P_{Fy,tire} - P_{conduction} + P_{ambient,tread} + P_{tread,road} \tag{14}$$

Power fluxes corresponding to the cooling flux due to the temperature difference between carcass and ambient $P_{ambient,carcass}$, and tire tread and ambient $P_{ambient,tread}$ are defined as:

$$P_{ambient,carcass} = h_{carcass}(T_{ambient} - T_{carcass})$$
$$P_{ambient,tread} = h_{tread}(T_{ambient} - T_{tread})$$

(15)

Power fluxes related to conduction between tread and carcass $P_{conduction}$ and between tread and road $P_{tread,road}$ are calculated as:

$$P_{conduction} = h_{conduction}(T_{tread} - T_{carcass})$$
$$P_{tread,road} = H_{tread-road}(T_{tread} - T_{road})$$

(16)

Compared to the Sorniotti model, the proposed thermal model due to a higher differential order should capture more dynamics related to heat flow and potentially produce better results.

The Kelly and Sharp model [18] states that in racing applications, the temperature of the tread significantly affects both the tire stiffness and the contact patch friction. It should be noted that the effect on the friction is higher. Since the rubber viscoelastic properties depend on temperature, the maximum performance on the racetrack is only available in a specific temperature range.

The tire model is also based on the brush model. Using the bristle stiffness $c_p$ the adhesion part can be presented:

$$c_p = \frac{w_{cp}G_{tread}}{h_{tread}}$$

(17)

where $w_{cp}$ is the contact patch width; $h_{tread}$ is the tread height.

The shear modulus of the tread $G_{tread}$ is defined as:

$$G_{tread} = \frac{G_{TA} - G_{\text{limit}}}{e^{-K_G T_{GA}}} e^{-K_G T_{tread}} + G_{\text{limit}}, \text{ where } K_G = \frac{\log(G_{TA} - G_{\text{limit}}) - \log(G_{TB} - G_{\text{limit}})}{T_{GB} - T_{GA}}$$

(18)

where $G_{TA}$ is the reference shear modulus at temperature A; $G_{TB}$ is the reference shear modulus at temperature B; $G_{limit}$ is the limit shear modulus value at a high temperature; $T_{GA}$ is the reference temperature A; $T_{GB}$ is the reference temperature B.

The sliding part of the contact patch is described using the dynamic friction coefficient. The master curve of the friction coefficient $\mu_{mc}$ for the tread compound is assumed to have a Gaussian shape and it describes a friction curve with a Gaussian shape on a log frequency axis:

$$\mu_{mc} = \mu_{base} + \left[\mu_{peak} - \mu_{base}\right] e^{-(K_{shape}(\log_{10} v_s - K_{shift}(T_{tread} - T_{ref})))^2}$$

(19)

where $\mu_{base}$ and $\mu_{peak}$ are the lower and upper limits of the dynamic friction coefficient; $K_{shape}$ is the master curve shape factor; $v_s$ is the sliding velocity; $K_{shift}$ is the master curve temperature shift factor; $T_{tread}$ is the tread temperature; $T_{ref}$ is the master curve reference temperature.

In addition, the model takes into account the influence of contact patch pressure on friction coefficient. It is assumed that friction coefficients are reduced linearly with contact patch pressure. The static and dynamic friction coefficients are defined as:

$$\mu_0 = K_{cpp}\mu_{ref}$$
$$\mu_d = K_{cpp}\mu_{mc}, \text{ where } K_{cpp} = 1 - K_{pcpp}\frac{p_{cp}}{K_{refcpp}}$$

(20)

where $p_{cp}$ is the contact patch pressure; $K_{cpp}$ is the parameter of friction reduction rate; $K_{pcpp}$ is the friction roll-off factor with contact patch pressure; $K_{refcpp}$ is the reference contact patch pressure; $\mu_0$ is the static friction coefficient; $\mu_d$ is the dynamic friction coefficient; $\mu_{ref}$ is the reference static friction coefficient.

Compared to the Kelly and Sharp model, the peak friction coefficient $\mu_{peak}$ is temperature-dependent in the proposed model, which could result in better accuracy.

### 3.3. Comparison Results

The simulation results obtained from the considered thermal models were compared with measurement data for various maneuvers. The initial values of coefficients were taken from [18,25] and further tuned using constrained parameter optimization.

To tune parameters, a non-linear least square method was applied to obtain a suitable approximation of temperature effect and was based on the error from the measurement defined as:

$$e_{meas} = \sqrt{\frac{\sum (T_{\text{model}} - T_{\text{measurement}})^2}{\sum T_{\text{model}}^2}} \tag{21}$$

It was checked that the fitting parameters provide a good estimation for different maneuvers. For example, the parameters were first optimized for lateral tests and later checked again for acceleration/braking tests. Then an average was taken for the tuning parameters to cover all the experimental tests and represent the whole driving envelope. The comparisons between the measured temperature, the temperature obtained from the Kelly and Sharp model, and the proposed thermal model are shown in Figure 7.

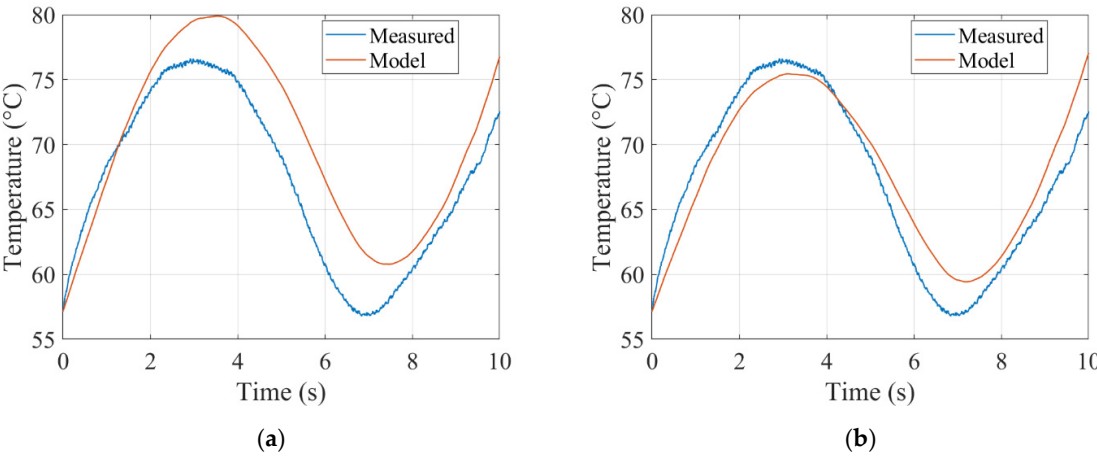

(a)                                                                 (b)

**Figure 7.** Temperature change in pure braking test: (**a**) results of the Kelly and Sharp model; (**b**) results of the proposed model.

The accumulated error for the compared thermal models is summarized in Table 2.

**Table 2.** Accumulated errors for compared thermal models.

|                    | **Sorniotti Model** | **Kelly and Sharp Model** | **Proposed Model** |
| ------------------ | ------------------- | ------------------------- | ------------------ |
| Pure acceleration  | 5.2%                | 4.7%                      | 5.5%               |
| Pure braking       | 2.5%                | 10%                       | 2.9%               |
| Pure cornering     | 3.2%                | 3.0%                      | 2.6%               |

The results of the proposed thermal model and the compared models are close to the measurement data for all conducted tests. The proposed thermal model provides the smallest error for pure cornering. For pure braking the Sorniotti model shows the best performance and for pure acceleration the smallest error can be achieved using the Kelly and Sharp model. However, the conclusions regarding adaptability of the models to different normal loads cannot be extracted from Table 2. Since the Kelly and Sharp model considers the contact patch area depending on the normal load and air pressure, the

model can adapt to different normal loads and the maximum error is limited to ~6%. Therefore, the models were also checked using measurement data for different load conditions. It was found that although the Sorniotti model has a good fit to the measured data, the error increases when the load condition is changed (Figure 8).

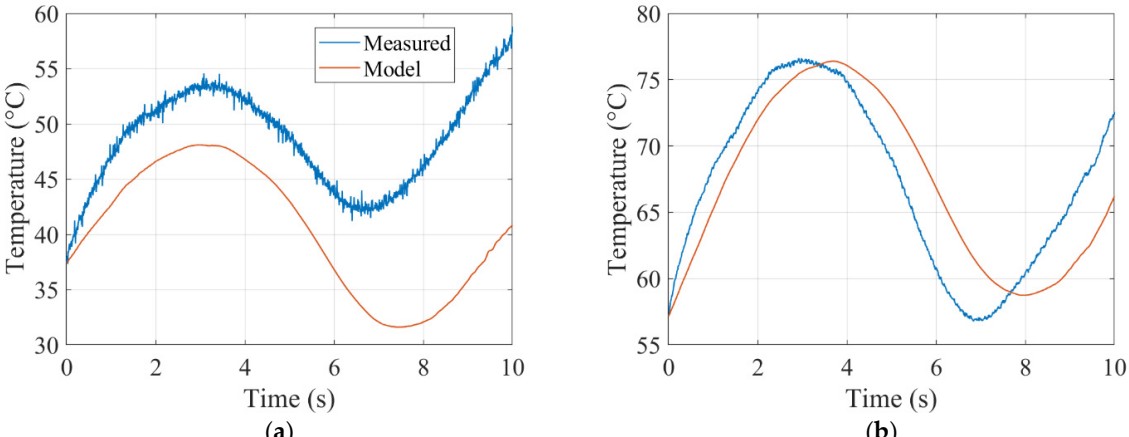

**Figure 8.** Temperature change in pure braking test using Sorniotti model: (**a**) normal load $F_z$ of 600 N; (**b**) normal load $F_z$ of 1000 N.

## 4. Modified Magic Formula Incorporating Temperature Effect

### 4.1. Baseline Parameter Estimation

Temperature plays a crucial role in tire performance. A large number of factors, such as the normal load, ambient conditions, etc., affect the temperature of the tread. Since the tread temperature profile was different on the two tires due to normal force fluctuation (two tires were tested simultaneously), it was necessary to align the measurements in such a way that the tires were in a similar temperature range.

Once the measurements were processed according to similar temperature ranges, the measurement data was converted according to TYDEX format [26]. This format enables the usage of MFTool software provided by TASS International (former TNO Automotive, Helmond, The Netherlands) [27]. The MFTool software is a commercial software to determine the coefficients of the empirical model, well known as the magic formula, from the pre-processed measurement data. It also allows to compensate for the variation in normal load and fits the curve for a nominal load. In the content of this study, MFTool was used to generate a tire property file for the extended magic formula (Appendix A).

### 4.2. Basic Magic Formula

The magic formula is an empirical model commonly used to simulate steady-state tire forces and moments. It is based on a special function instead of look-up tables or other various polynomial functions. By adjusting the function coefficients, the same special function can be used to describe longitudinal and lateral forces (sine function), and self-aligning moment (cosine function). The basic magic formula has the following form [28]:

$$
\begin{aligned}
y(x) &= D\sin(C\arctan Bx - E(Bx - \arctan Bx)) \\
Y(x) &= y(x) + S_v \\
x &= X + S_h
\end{aligned}
\tag{22}
$$

where $Y(x)$ may represent longitudinal, lateral forces or self-aligning moment; $X$ denotes slip angle $\alpha$ or longitudinal wheel slip $\kappa$; $B$ is stiffness factor; $C$ is shape factor; $D$ is peak factor; $E$ is curvature factor; $S_h$ is horizontal shift; $S_v$ is vertical shift.

The peak value is described by the peak factor *D*. The derivative at the origin provides *BCD* as a result and defines the longitudinal or cornering stiffness of the tire. The curve shape can be adjusted by both factors *C* and *E*, where *C*, called as shape factor, controls the "stretching" in the *x* direction and *E*, called the curvature factor, enables a local stretch or compression. The parameters $S_h$ and $S_v$ allow to model the influence of plysteer and conicity. The extended version of the magic formula [29] also covers effects of load dependency, combined slip, camber angle, inflation pressure, etc.

### 4.3. Proposed Modification for Temperature Effect

To change the peak friction and the stiffness of the force curve according to temperature variation, the magic formula was modified to affect the tire forces and moments. The proposed modifications are only demonstrated for the basic magic formula; meanwhile, they were integrated with the extended version of the magic formula. To represent temperature change, a delta temperature $\Delta T$ was defined:

$$\Delta T = \frac{T - T_{ref}}{T_{ref}} \tag{23}$$

where $T_{ref}$ is reference temperature (the performance is known from measurements).

To evaluate the performance of the proposed modification, the normalized friction values from measurement data and simulation results were compared. The measured temperature was used as an input to the modified magic formula.

Longitudinal Force. Four temperature coefficients were introduced to capture the temperature effect on the longitudinal force. The coefficients related to temperature effect are:

| | |
|---|---|
| $T_{X1}$ | Linear temperature effect on longitudinal slip stiffness |
| $T_{X2}$ | Quadratic temperature effect on longitudinal slip stiffness |
| $T_{X3}$ | Linear temperature effect on longitudinal friction |
| $T_{X4}$ | Quadratic temperature effect on longitudinal friction |

To affect the longitudinal slip stiffness and longitudinal friction, the above-mentioned coefficients were used. The modified peak factor $D_x$ and longitudinal slip stiffness $K_x$ are:

$$\begin{aligned} D_x &= \left(1 + T_{X3}\Delta T + T_{X4}\Delta T^2\right)D_{xbase} \\ K_x &= \left(1 + T_{X1}\Delta T + T_{X2}\Delta T^2\right)K_{xbase} \end{aligned} \tag{24}$$

where $D_{xbase}$ is the peak factor of pure longitudinal force *D* and $K_{xbase}$ is the longitudinal slip stiffness $K_x$ according to the basic magic formula.

As can be observed in Figure 9, the measurement data and simulation results of normalized longitudinal friction are close to each other. The quantitative comparison between simulation and experimental results for longitudinal force is summarized in Table 3. It also can be noted that temperature changes longitudinal stiffness and the longitudinal friction similar to experimental data. The main difference is related to the measurement noise and fluctuations due to road surface irregularities.

**Table 3.** Errors from measurements for different temperatures.

| Temperature Range | Longitudinal Force $F_x$ | Lateral Force $F_y$ |
|---|---|---|
| $T < 50\,°C$ | - | 18.35% |
| $50\,°C \leq T < 60\,°C$ | 14.35% | 11.53% |
| $60\,°C \leq T < 70\,°C$ | 12.9% | 4.9% |
| $70\,°C \leq T < 80\,°C$ | 12.5% | - |
| $80\,°C \leq T < 90\,°C$ | 10.4% | - |
| $T > 90\,°C$ | 10% | - |

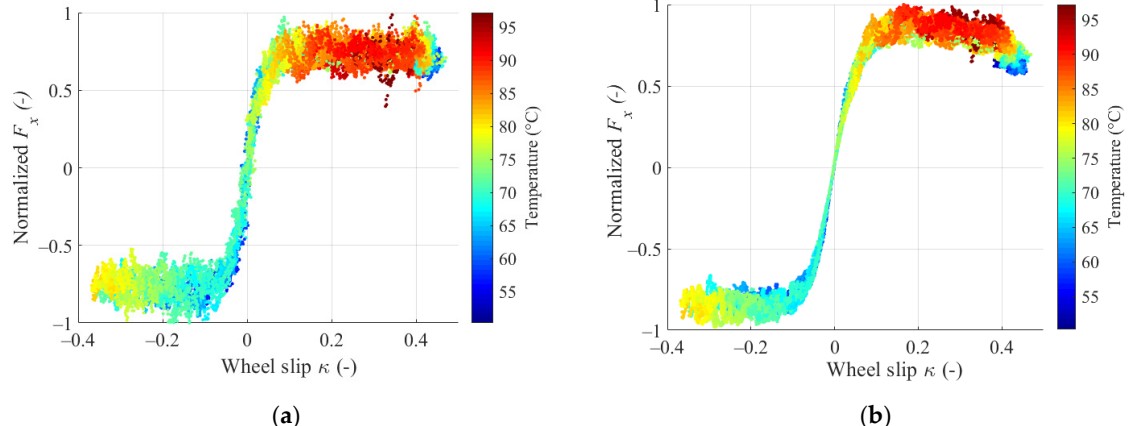

**Figure 9.** Normalized longitudinal force $F_x$: (**a**) experimental measurements; (**b**) modified magic formula.

Lateral Force and Self-Aligning Moment. Four temperature coefficients were introduced to incorporate the temperature effect in lateral force variation:

$T_{Y1}$     Temperature effect on cornering stiffness magnitude
$T_{Y2}$     Temperature effect on location of cornering stiffness peak
$T_{Y3}$     Linear temperature effect on lateral friction
$T_{Y4}$     Quadratic temperature effect on lateral friction

To affect the cornering stiffness and lateral friction respectively, the above-mentioned coefficients were used. The modified peak factor $D_y$ and the cornering stiffness $K_y$ are described:

$$
\begin{aligned}
D_y &= \left(1 + T_{Y3}\Delta T + T_{Y4}\Delta T^2\right)D_{ybase} \\
K_y &= (1 + T_{Y1}\Delta T)P_{KY1}F_{z0}\sin\left(\arctan\left(\frac{F_z}{P_{KY2}F_{z0}(1+T_{Y2}\Delta T)}\right)\right)
\end{aligned}
\tag{25}
$$

where $D_{ybase}$ is the peak factor of pure lateral force $D$ in the basic magic formula; $F_{z0}$ is the nominal normal load; $P_{KY1}$ is the maximum value of stiffness $K_{fy}/F_{z0}$; $P_{KY2}$ is the load at which $K_{fy}$ reaches maximum value.

The results of normalized lateral force from measurement data and simulation results are shown in Figure 10 and demonstrate a similar behavior while temperature changes. The quantitative assessment between simulation and experimental results for lateral force is summarized in Table 3. The pneumatic trail would remain constant under the assumption of the same pressure distribution in a contact patch. Since the self-aligning moment is the product between the lateral force and the tire pneumatic trail, it will change only due to the lateral force variation. Therefore, there is no need to introduce the coefficients affecting the self-aligning moment as shown in Figure 11.

The error to compare the performance of the modified magic formula is defined as:

$$
e_{meas} = 100\%\frac{\sqrt{(F_{simulation} - F_{measurement})^2}}{|F_{measurement}|}
\tag{26}
$$

Data for different temperature ranges were taken to evaluate the error values for the longitudinal and lateral forces, and the summary is presented in Table 3.

As it can be observed, the modified magic formula performs sufficiently close to the actual measurement data taking into account that the absolute measurement error was 5%. The largest errors occur at lower temperatures and a general trend of improved accuracy is seen as the temperature increases. For lower temperatures, the potential reason of the deviation is related to a high variance of

measured data. The errors for the self-aligning moment are not shown due to the high variance in the measurement data related to the measurement setup.

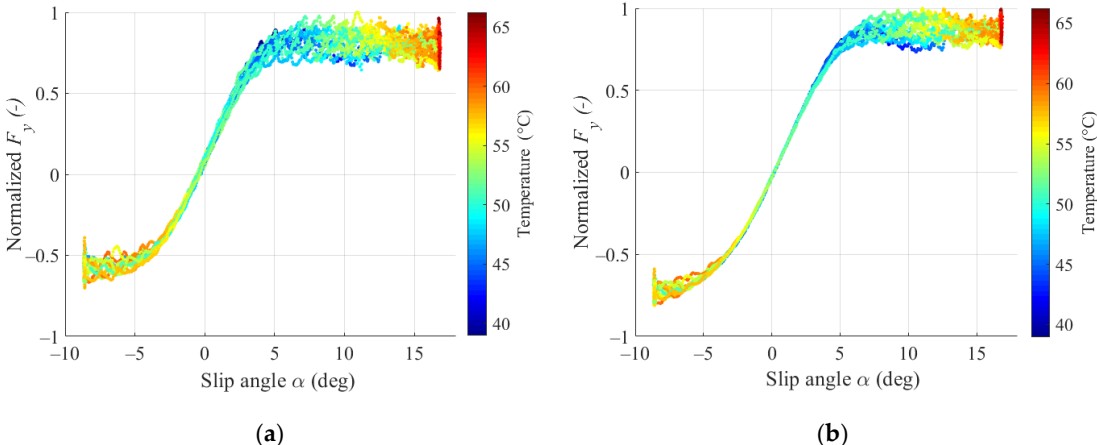

**Figure 10.** Normalized lateral force $F_y$: (**a**) experimental measurements; (**b**) modified magic formula.

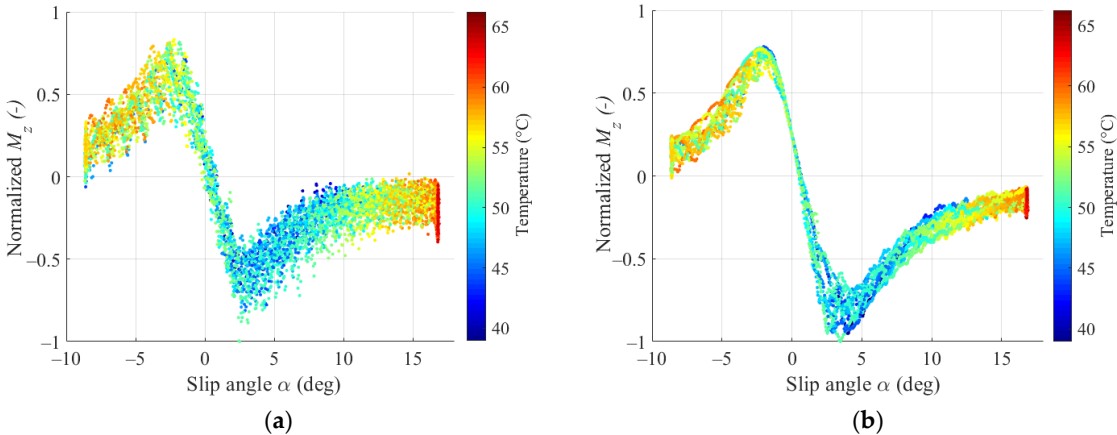

**Figure 11.** Normalized self-aligning moment $M_z$: (**a**) experimental measurements; (**b**) modified magic formula.

## 5. Integration into Full FSAE Vehicle Model

### 5.1. Full Vehicle Modeling

The following assumptions have been made for the multi-body model of the FSAE vehicle:

- All joints and bodies besides the tires and springs are considered as rigid. Since the suspension and chassis elements are designed to be stiff, compliance is neglected. Joint friction is omitted, except modeling of the transmission friction. For this purpose, damping in the revolute joints between the suspension and the wheels is added.
- Aerodynamic forces are applied on a constant center of pressure. Its position was calculated based on computational fluid dynamic simulations in steady-state conditions and changes according to the velocity, roll, pitch, etc.
- The characteristics of suspension springs and dampers are linear.

The model was developed in Simscape Multibody (Simulink toolbox). The coordinate system and axis orientation was according to the SAE coordinate system. Center of gravity position and mass-inertial characteristics of the chassis were obtained from the computer-aided design (CAD) model or/and actual measurements on the FSAE vehicle. The model includes four suspension subsystems

connected to the chassis body. Using revolute joints from the upright center, the tire was connected to the suspension. The suspension geometry was imported from the CAD model corresponding to the actual FSAE vehicle. A steering system was connected to the front suspension and the chassis body through different hardpoints. More details regarding the vehicle modeling can be found in [30].

## 5.2. Model Structure

The completed structure of the simulation setup is shown in Figure 12. The tire states (position, velocity, acceleration) are calculated on each time step based on the multi-body vehicle model and transferred to the modified magic formula. According to kinematic states, the magic formula calculates the forces and moments. Then tire forces and moments are sent to the thermal model calculating the temperature and its change. Finally, tire forces and moments are transferred to the multibody vehicle model, while temperature and pressure are transferred to the tire model.

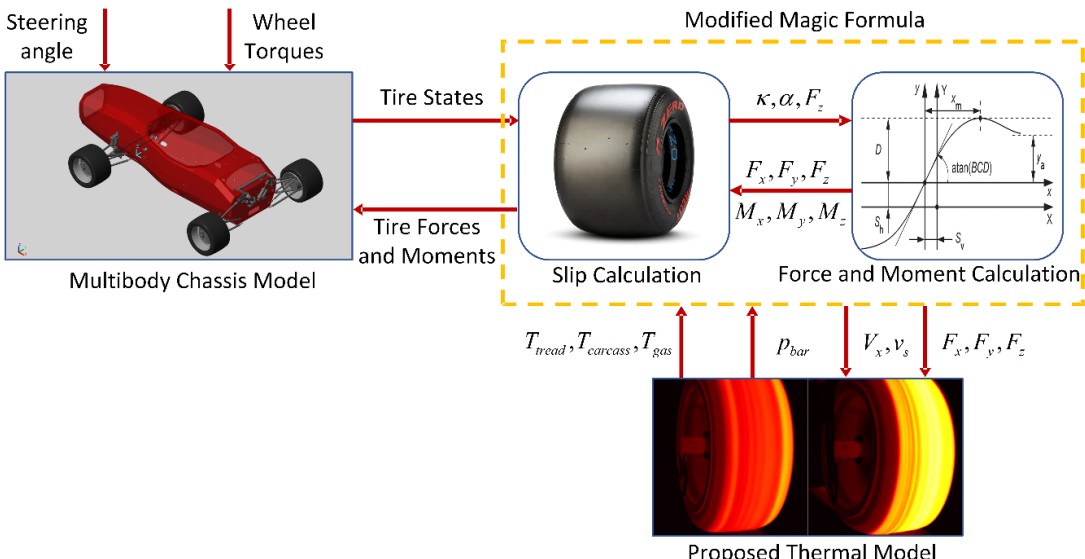

**Figure 12.** Model structure.

## 5.3. Validation

To validate the overall model, two test scenarios were considered. The first test scenario was a skidpad maneuver to evaluate a steady-state lateral behavior. The second test scenario was a full lap maneuver performed for a longer time.

Skidpad Maneuver. The skidpad course consists of two pairs of concentric circles in a figure of eight pattern. The centers of these circles are 18.25 m apart. The inner circles are 15.25 m in diameter and the outer circles are 21.25 m in diameter. The driving path is the 3 m wide path between the inner and outer circles [31].

To increase the signal-to-noise ratio without significant signal distortion, the raw sensor data was filtered using a Savitzky–Golay digital filter [32]. The filtered data was used for comparison with simulation results. The longitudinal and lateral accelerations, and yaw rate are shown in Figure 13.

The simulation shows more oversteered behavior compared to the measured data. The mismatch can be caused by the modeling of chassis roll stiffness. A real chassis has a limited roll stiffness; however, as mentioned, the chassis was assumed as a rigid body. Hence, the chassis was infinitely stiff in the simulation model. It could result in a higher yaw rate and increase the root mean square error (RMSE) of the yaw rate.

The longitudinal and lateral accelerations, and yaw rate were better aligned with the measurement data in the left turn circles. This observation can be explained by the fact that the elevation change in the skidpad circuit affected the measurements from the accelerometer sensor.

Figure 14 shows the evolution of tire temperature during skidpad maneuver for four wheels: front left (FL), front right (FR), rear left (RL), and rear right (RR). As can be observed, due to higher normal loads, the outer tires (initially left tires for the right turn) start heating up first. When the vehicle steered in another direction (at 15 s), the right tires start heating up.

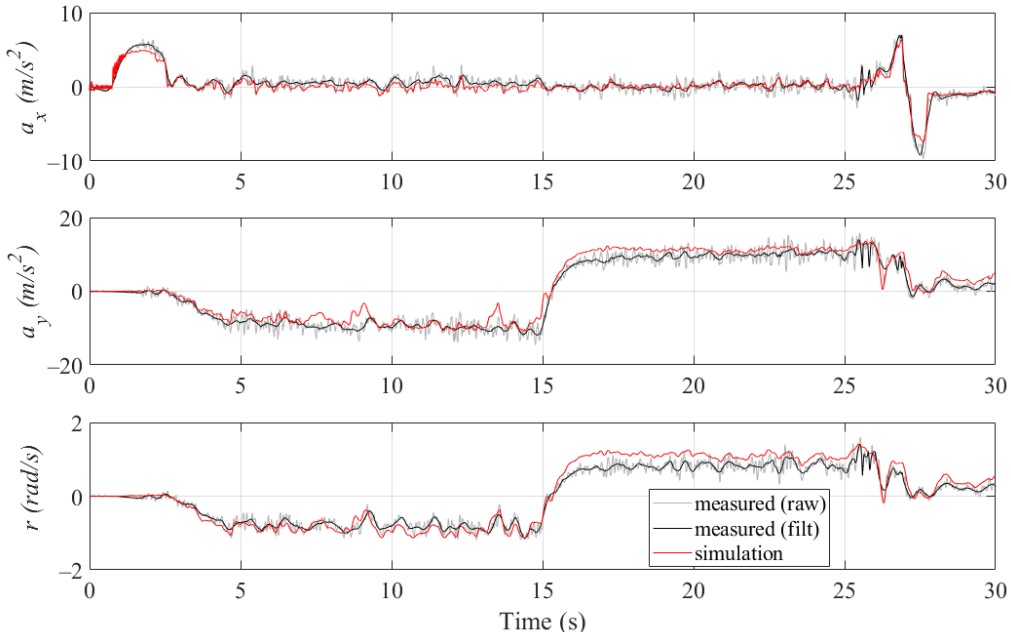

**Figure 13.** Longitudinal acceleration $a_x$, lateral acceleration $a_y$, and yaw rate $r$ for skidpad.

Full Lap. To validate transient behavior of the simulation model, a full lap maneuver was investigated. The measurement data correspond to one of the endurance runs with a duration of ~22 km driving (18–20 laps). The simulation was conducted for one lap of the circuit performed approx. in 50 s.

In a similar way as was done for the skidpad scenario, the raw measurement data of longitudinal and lateral accelerations, and yaw rate, were filtered and the RMSE calculated. The RMSE for the longitudinal and lateral accelerations, as well as yaw rate are summarized in Table 4, including also the simulation results performed without the proposed model. The comparison between simulation results with a temperature effect and measurement data is shown in Figure 15. It can be noted that the yaw rate from the simulation results was aligned with the measured data; however, an offset can be observed which might relate to the above-mentioned difference in the chassis roll stiffness.

**Table 4.** The root mean square error (RMSE) for both simulation maneuvers.

| | Skidpad Maneuver | | Full Lap Maneuver | |
|---|---|---|---|---|
| Temperature effect | without | with | without | with |
| Longitudinal acceleration, m/s$^2$ | 0.62 | 0.65 | 1.50 | 1.42 |
| Lateral acceleration, m/s$^2$ | 1.86 | 1.26 | 4.77 | 3.19 |
| Yaw rate, rad/s | 0.27 | 0.16 | 0.37 | 0.16 |

The RMSE for longitudinal acceleration is slightly higher compared to the skidpad scenario; however, the longitudinal acceleration in the skidpad maneuver was much lower compared to a full lap scenario. Thus, the simulation results of longitudinal dynamics using wheel torques as an input matched the longitudinal acceleration as sufficiently accurate.

The highest difference was found in lateral dynamics by the RMSE of lateral acceleration of 3.19 m/s$^2$. The potential reasons can be related to: (i) combined slip conditions; (ii) camber effect on lateral

force, (iii) longitudinal velocity not tacked precisely. Due to a limited financial budget, the tire testing was performed only for pure longitudinal and lateral conditions. The coefficients corresponding to combined slip behavior were estimated based on similarly sized FS tires. It can be observed, that the maximum mismatch happens in a corner where the tires were cambered due to chassis roll and performed in combined slip conditions. Another aspect is the body slip of the actual vehicle which might affect the differences between the simulation and real FSAE vehicles.

Figure 16 shows the evolution of tire temperature during a full lap maneuver. The tread temperature steadily increased and its reduction corresponds to straight lines or long corners. It can also be observed that a higher temperature for right tires was achieved compared to the left tires. Due to the circuit orientation, the right tires have higher loads. In this case, the gas temperature increased and due to the deflections, the carcass heated up.

As can be concluded from Table 4, the RMSE of lateral acceleration was up to 30% and 33% in steady-state and transient maneuvers, correspondingly. Regarding the yaw rate, the RMSE was up to 41% in steady-state maneuver and up to 56% in the transient one. Therefore, the modeling of temperature effect has a significant effect on FSAE vehicle dynamics.

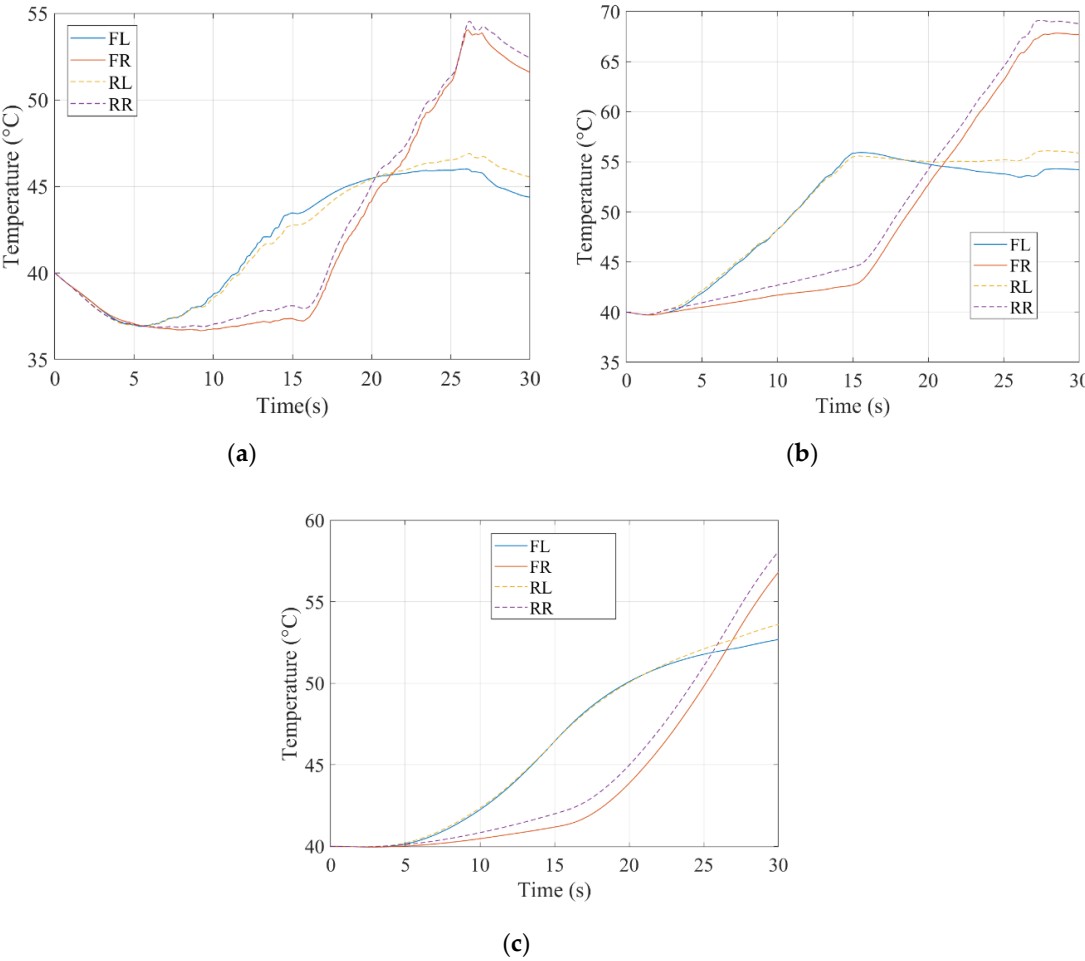

**Figure 14.** Evolution of tire temperatures for skidpad maneuver: (**a**) tire tread temperature; (**b**) tire carcass temperature; (**c**) tire inflation gas temperature.

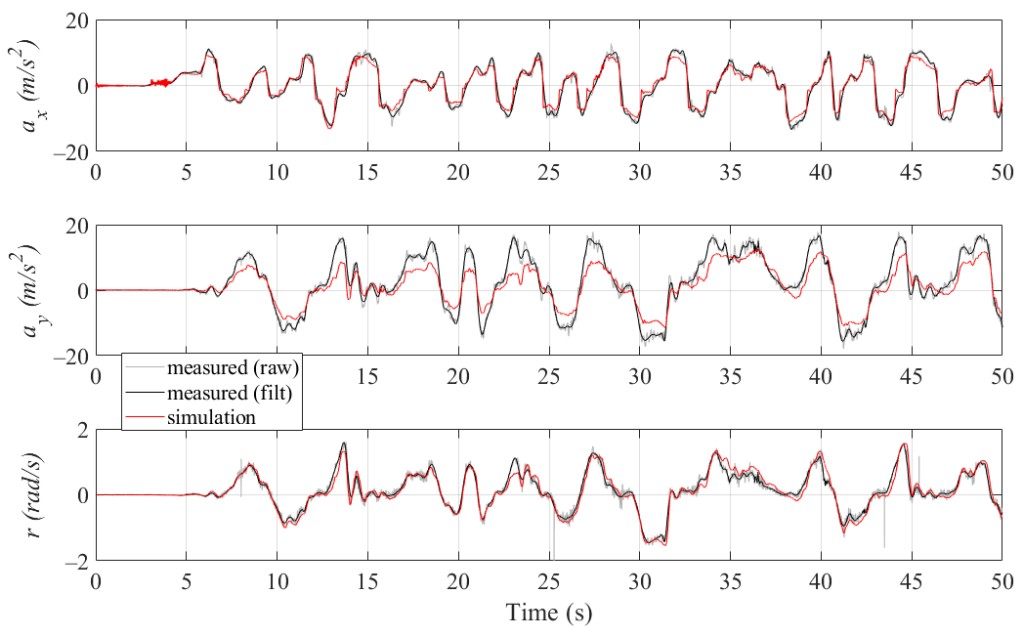

**Figure 15.** Longitudinal acceleration $a_x$, lateral acceleration $a_y$, and yaw rate $r$ for a full lap.

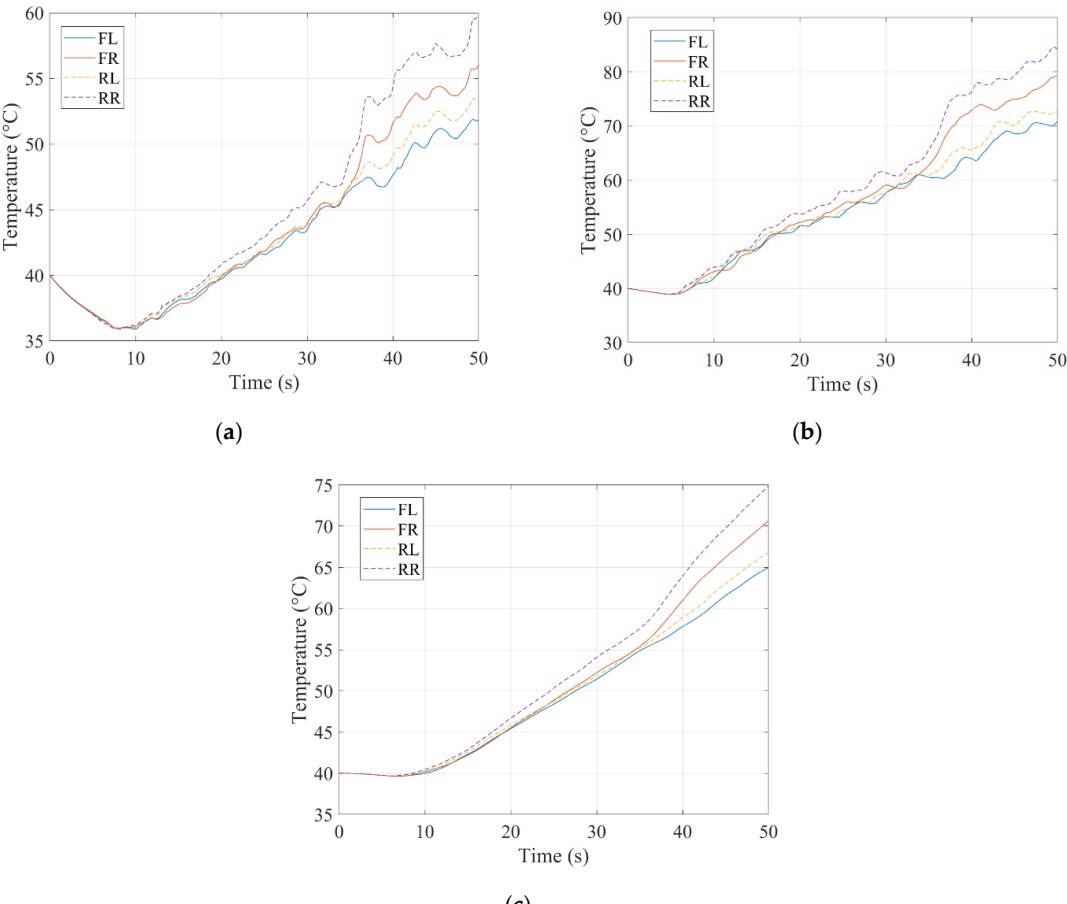

**Figure 16.** Evolution of tire temperatures for full lap: (**a**) tire tread temperature; (**b**) tire carcass temperature; (**c**) tire inflation gas temperature.

## 6. Conclusions

This paper proposes an empirical tire model (modified magic formula) coupled with a thermal model to incorporate temperature effects on the force capability of the tire. The thermal model was proposed to model the influence of temperature on peak friction and the shear modulus, allowing for a better representation of the temperature variation under different load conditions. The magic formula was modified introducing new coefficients allowing for the change in longitudinal and cornering stiffness, and peak factors. The simulation results of the modified magic formula with temperature sufficiently matched the variation in longitudinal and lateral force, as well as self-aligning moment, compared to the experimental tire tests. The error between simulation results and experimental data was in the range between 10% and 15%. However, limited accuracy with an error of 18.5% was achieved at lower temperatures due to a high variance in the measured data.

To evaluate the proposed tire model, the thermal and modified magic formula models were integrated to a multi-body vehicle model. The simulation results match the behavior of the real FSAE vehicle in steady-state maneuver. For transient maneuvers, limited accuracy was achieved due to estimated parameters or unmodeled dynamics such as tire behavior in combined slip conditions, neglect of camber effect on lateral force, etc. Comparing the proposed model with an empirical-based tire model without temperature effect, the RMSE of lateral acceleration was reduced by up to 30% and 33% in the steady-state and the transient maneuvers, correspondingly. For yaw motion, the RMSE of the yaw rate was reduced up to 41% in the steady-state maneuver and up to 56% in the transient one.

Further work related to tire testing which would result in a more precise identification of the magic formula parameters and thus improve the tire model. The following improvements should be considered: (i) combined slip conditions, (ii) camber influence on performance, and (iii) relaxation behavior.

**Author Contributions:** Conceptualization, D.H. and B.S.; testing, D.H.; modeling, D.H.; supervision, B.S.; validation, D.H.; writing—original draft, D.H.; writing—review and editing, B.S.

**Funding:** This research received no external funding.

**Acknowledgments:** The authors would like to thank Formula Student Team Delft, The Netherlands, for their support with tire and vehicle testing.

**Conflicts of Interest:** The authors declare no conflicts of interest.

## Appendix A —Tire Property File

```
[MODEL]
FITTYP                =62                $Magic Formula Version number
[DIMENSION]
UNLOADED_RADIUS       =0.17              $Free tyre radius
WIDTH                 =0.2165            $Nominal section width of the tyre
RIM_RADIUS            =0.127             $Nominal rim radius
RIM_WIDTH             =0.225             $Rim width
ASPECT_RATIO          =0.1986            $Nominal aspect ratio
[INERTIA]
MASS                  =2.825             $Tyre mass
IXX                   =0.0538            $Tyre diametral moment of inertia
IYY                   =0.1               $Tyre polar moment of inertia
BELT_MASS             =2.2               $Belt mass
BELT_IXX              =0.05              $Belt diametral moment of inertia
BELT_IYY              =0.08              $Belt polar moment of inertia
GRAVITY               =−9.81             $Gravity acting on belt in Z direction
```

[VERTICAL]
FNOMIN                     =600               $Nominal wheel load
VERTICAL_STIFFNESS         =95000             $Tyre vertical stiffness
VERTICAL_DAMPING           =50                $Tyre vertical damping
BREFF                      =8.4               $Low load stiffness e.r.r.
DREFF                      =0.27              $Peak value of e.r.r.
FREFF                      =0.07              $High load stiffness e.r.r.
Q_RE0                      =0.9974            $Ratio of free tyre radius with nominal tyre radius
[LONGITUDINAL_COEFFICIENTS]
PCX1                       =1.391             $Shape factor Cfx for longitudinal force
PDX1                       =1.5314            $Longitudinal friction Mux at Fznom
PDX2                       =−0.04906          $Variation of friction Mux with load
PEX1                       =0.4454            $Longitudinal curvature Efx at Fznom
PEX2                       =0.2192            $Variation of curvature Efx with load
PEX4                       =0.1665            $Factor in curvature Efx while driving
PKX1                       =43.63             $Longitudinal slip stiffness Kfx/Fz at Fznom
PKX2                       =4.4735            $Variation of slip stiffness Kfx/Fz with load
PKX3                       =0.023027          $Exponent in slip stiffness Kfx/Fz with load
PHX1                       =−0.003839         $Horizontal shift Shx at Fznom
PHX2                       =0.0044605         $Variation of shift Shx with load
PVX1                       =0.04359           $Vertical shift Svx/Fz at Fznom
PVX2                       =0.007515          $Variation of shift Svx/Fz with load
RBX1                       =10                $Slope factor for combined slip Fx reduction
RBX2                       =6                 $Variation of slope Fx reduction with kappa
RCX1                       =1                 $Shape factor for combined slip Fx reduction
[LATERAL_COEFFICIENTS]
PCY1                       =1.3318            $Shape factor Cfy for lateral forces
PDY1                       =1.6502            $Lateral friction Muy
PDY2                       =−0.14737          $Variation of friction Muy with load
PEY1                       =0.5               $Lateral curvature Efy at Fznom
PEY2                       =−9.1214E-7        $Variation of curvature Efy with load
PKY1                       =−85               $Maximum value of stiffness Kfy/Fznom
PKY2                       =5                 $Load at which Kfy reaches maximum value
PKY4                       =1.7923            $Curvature of stiffness Kfy
PHY1                       =0.008             $Horizontal shift Shy at Fznom
PVY1                       =0.1               $Vertical shift in Svy/Fz at Fznom
PVY2                       =−0.001143         $Variation of shift Svy/Fz with load
RBY1                       =16                $Slope factor for combined Fy reduction
RCY1                       =1                 $Shape factor for combined Fy reduction
[ALIGNING_COEFFICIENTS]
QBZ1                       =7                 $Trail slope factor for trail Bpt at Fznom
QBZ2                       =2                 $Variation of slope Bpt with load
QCZ1                       =1.2               $Shape factor Cpt for pneumatic trail
QDZ1                       =0.12              $Peak trail Dpt = Dpt*(Fz/Fznom*R0)
QDZ2                       =−0.02             $Variation of peak Dpt with load
QEZ1                       =−2.8              $Trail curvature Ept at Fznom
QEZ2                       =3                 $Variation of curvature Ept with load

```
[TEMPERATURE_COEFFICIENTS]
TY1                    =−0.25           $Temperature effect on cornering stiffness magnitude
TY2                    =0.15            $Temperature effect on location of cornering stiffness peak
TY3                    =0.25            $Linear temperature effect on lateral friction
TY4                    =−0.1            $Quadratic temperature effect on lateral friction
TX1                    =−0.25           $Linear temperature effect on slip stiffness
TX2                    =0.15            $Quadratic temperature effect on slip stiffness
TX3                    =0.25            $Linear temperature effect on longitudinal friction
TX4                    =−0.1            $Quadratic temperature effect on longitudinal friction
TREF                   =50              $Reference temperature
```
Note: the non-mentioned coefficients are equal to zero.

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
