# Peer review of "Tire Model with Temperature Effects for Formula SAE Vehicle"

_applsci, doi:10.3390/app9245328_

Round 1

Reviewer 1 Report

The paper discusses the extension of the regular tire friction model (e.g., magic formula) to consider the temperature influence on friction force characteristics of tire-ground interactions and then uses it for Formula Student (FS) car simulation studies. The paper is clearly written and easy to follow. The approach sounds reasonable. I have the following comments and suggestions.

The paper contains some interesting and engineering approaches to estimate the influence of tire temperature on friction force properties. I felt that the authors should try to strengthen the technical depth and discussion and further brings this work into a scientific publication. For example, it is not well-justified why the Magic formula is used and why the temperature influence only impacts on specific model parameters such as D and E, not for others. Second, I would suggest the authors clearly state the contribution of the work given the existing similar publication in literature. This should be placed in the introduction section along with literature review. Finally, the authors might consider the focus of the work not only for FS car and might be helpful and useful for other general passenger vehicles. At least some discussions for this is helpful to generalize the contribution of the work. It would be good to give more high-fidelity simulation results or experiments using the extended tire friction model to demonstrate the significance of the work. The authors might at least discuss this. As a scientific publication, I would suggest the authors discuss and review a bit about tire friction models and explain why the magic formula is the selection to conduct model extension. Some other models such as LuGre friction model or brush tire model can have advantages of physical meaning and interpretation for physical variations such as temperature change. For example, the work in Li et al. "A hybrid physical-dynamic tire/road friction model" in ASME Journal of Dynamic Systems, Measurement and Control, vol. 135, no. 1, article 011007, gives a review of other tire models and the authors might consider to cite a few related work in tire friction models and discuss possible model extension.  

Author Response

The response is attached (Word document).

Reviewer 2 Report

The paper has proposed a thermal model for the FSAE vehicle tire. The proposed thermal model of the tire were compared with two existing model: Kelly & Sharp model and Sorniotti model. Then an extended Magic Formula (original one used for steady-state tire force and moments) with the thermal consideration were integrated into a full FSAE vehicle model. Most simulation results of the thermal model were matched with their measurements but the dynamic results has limited accuracy. All simulation results are based on the parameters optimized from the measurement. Also section 4 and 5 had no cross-compression with other thermal model. Thus it is hard to say the model is suitable for general case or just for your special case. 

Comments:

Figure 5, please indicted the test condition such as normal loads and tire types

Equation 1 considered the heat transformation in the system with only two heat source into the system, the Qsliding and Qdamp. Is there any other heat sources, such as heat from the engine system and so on?

Figure 6, please indicate clear of the two figures with caption, which one is measurement data?

Page 7 , “For a detailed description of the Sorniotti model we refer the reader to [19].” is not a good present way for journal paper. So does page 8, “For a detailed description of the Kelly & Sharp model we refer the reader to [13].”

Page 8, “It is assumed that friction coefficients are reduced linearly with contact patch pressure.” Is this assumption from any reference?

Equation 20, Acp is not appeared in the equation; what is ud and umc presents?

Equation 21, what is subscript “meas” mean? “Measurement data”?

From Table 2. It is showing that the proposed model has lese error in pure cornering test. However, the fitting parameters were optimized for the lateral tests and other two results of the proposed model is no better than other two model. Figure 7 and figure 8 looks be the special cases of the two model: biggest error in pure braking for the Kelly & Sharp model from Table 2 and special cases of and Sorniotti model (how about your model?). The simulation results may not enough to prove that the proposed model are better than other two in general.

Section 4.3, the simulation results are close to the measurement results. However, how it improves on the results from the basic Magic Formula? And the same problem, the simulation results are based on the parameters from your measurement, which may not so strong to prove that your model is accurate and better.

Section 5, the integration into the full FSAE vehicle model only compared with model without temperature effect. How is the proposed model comparing to other model with temperature consideration? Or ever the measurement data?

The present of the paper is not well organized. Section 3 and 4 are not well related until section 5.2.

Author Response

(The authors gave the same response as above.)

Reviewer 3 Report

Please find remarks in the attached file

Author Response

(The authors gave the same response as above.)

Round 2

Reviewer 2 Report

The manuscript has been significantly improved with all comments responded.